# A Survey on Optimization Techniques for Edge Artificial Intelligence (AI)

**DOI:** 10.3390/s23031279

**Published:** 2023-01-22

**Authors:** Chellammal Surianarayanan, John Jeyasekaran Lawrence, Pethuru Raj Chelliah, Edmond Prakash, Chaminda Hewage

**Affiliations:** 1Centre for Distance and Online Education, Bharathidasan University, Tiruchirappalli 620024, Tamilnadu, India; 2Cardiff School of Technologies, Cardiff Metropolitan University, Cardiff CF5 2YB, UK; 3Edge AI Division, Reliance Jio Platforms Ltd., Bangalore 560103, Karnataka, India; 4Research Center for Creative Arts, University for the Creative Arts (UCA), Farnham GU9 7DS, UK

**Keywords:** artificial intelligence, AI model optimization, edge AI, federated learning, optimization methods for edge AI, energy efficient methods for edge AI

## Abstract

Artificial Intelligence (Al) models are being produced and used to solve a variety of current and future business and technical problems. Therefore, AI model engineering processes, platforms, and products are acquiring special significance across industry verticals. For achieving deeper automation, the number of data features being used while generating highly promising and productive AI models is numerous, and hence the resulting AI models are bulky. Such heavyweight models consume a lot of computation, storage, networking, and energy resources. On the other side, increasingly, AI models are being deployed in IoT devices to ensure real-time knowledge discovery and dissemination. Real-time insights are of paramount importance in producing and releasing real-time and intelligent services and applications. Thus, edge intelligence through on-device data processing has laid down a stimulating foundation for real-time intelligent enterprises and environments. With these emerging requirements, the focus turned towards unearthing competent and cognitive techniques for maximally compressing huge AI models without sacrificing AI model performance. Therefore, AI researchers have come up with a number of powerful optimization techniques and tools to optimize AI models. This paper is to dig deep and describe all kinds of model optimization at different levels and layers. Having learned the optimization methods, this work has highlighted the importance of having an enabling AI model optimization framework.

## 1. Introduction

The Internet of Things (IoT) has grown rapidly and generates a huge amount of data. Depending upon the domain and application, say, for example, smart traffic application, smart home, smart city, smart transport, etc., the acquired data are required to be processed immediately to produce meaningful insights and actionable decisions. In these cases, sending data to a centralized server and analyzing the data at the server involves greater latency [1] which even prohibits the real purpose of the application itself. Cloud computing is not adequate to meet the diverse needs of data analysis of today’s intelligent society, and so edge computing has evolved [2,3]. Edge computing has brought the processing of data to the point of acquisition by pushing applications, storage, and processing power away from the centralized data center and to the edge itself [4].

Centralized processing requires a massive amount of need to be transferred to the cloud for analysis. This not only requires more network bandwidth but also consumes time. Thus, it seriously suffers from latency, bandwidth-related issues, and huge transmission energy, which cannot be tolerated in applications involving augmented reality, video conferences, streaming applications, etc. However, in reality, every network has limited bandwidth. In addition, when data are transmitted to the cloud, it inherently prohibits the real-time analysis of data. However, many applications are in need of real-time analysis. For example, in the case of the healthcare domain, consider the case of a vital parameter-monitoring system that monitors parameters related to, say, COVID-19. Despite sending the monitored data to a centralized cloud server to which the physician or hospital is connected, which upon receiving the data, will analyze the health conditions of the patient, the monitoring system itself should be equipped with processing capability so that it can produce actionable and meaningful insights without latency. In addition, it itself will direct the required action and prevents the transfer of data to a centralized infrastructure. In scenarios where the analysis in edge becomes necessary, it should be conducted at the edge itself without centralized analytics.

Further, the need for analysis at edge devices becomes very crucial in applications involving augmented reality, video conference, streaming applications, games, Content Delivery Networks (CDN), remote robotics-based surgery, product inspection, preventive maintenance, autonomous car, Industrial Internet of Things, smart power distribution, security related applications are in a compelling need of analysis at the edge. Another important aspect is with respect to the security and privacy of the data. In certain applications, such as the financial domain, the data at the edge needs to be kept in itself to maintain security and privacy. Moreover, in certain other applications, data optimization is required. In a nutshell, several applications are in need of analysis of data near the data source itself.

Having understood the need for analysis at the edge, the next question comes to one’s mind is that how to process them. Obviously, the amount of data involved in many domains is large, and it keeps on becoming massive. So, to handle massive data efficiently and automatically in less time, Artificial Intelligence (AI) based algorithms are mandatorily required. However, the creation, training, and execution of AI algorithms require substantial processing power, storage, and a huge amount of data for learning. In particular, the recent introduction of deep neural networks consists of hundreds of billions of parameters that require high processing power and storage. Two aspects, namely, training and inference of the models to be considered seriously. First, an AI model needs to be trained by a labeled dataset. Bigger datasets for the training phase cause explosive growth in processing, storage, and energy consumption. In addition, training takes place initially or intermittently. Once the model is trained, it’s ready to find meaningful patterns in new data. This process is called inference. The inference is a continuous activity, and it involves fewer data but consumes energy continuously.

Despite the compelling needs of Al in edge devices, training and executing AI models on edge devices is really challenging due to the following reasons

Edge devices are limited in their hardware capabilities with respect to processing power and memory, network bandwidthThe devices are limited network bandwidth.Edge devices are often battery-operated and low-power devicesThe resources of edge devices do not scale as cloud resources, and edge resources are heterogeneous, which may degrade the service quality [5]Accomplishing collaboration among heterogeneous devices is difficult [6]Original private data cannot be used for model optimization. Distribution of computation to devices is also limited by limited communication resources of the devices [7]

AI models have the potential for huge carbon footprints. In order to make them energy efficient, it is necessary to bring in a stream of optimizations in hardware, software, and data usage. Architectures for artificial intelligence in combination with the Internet of Things are required to be established. The training and inference phases of AI models require the cooperation of specialized hardware designs, appropriate architecture selection and model optimization, and other optimization methods [8].

In this context, the research question of this review paper is to analyze the publications relevant to optimization methods for artificial intelligence-based models to be trained and executed in edge computing environments and present the meaningful inference drawn from the analysis in a categorical manner so that the readers can perceive the existing methods with more clarity. Thus, the scope of the work is set to investigate the publications relevant to optimization methods of AI models for edge computing.

The organization of the paper is as follows. Section 2 highlights various edge AI use cases. Section 3 briefly describes the method used for this survey. Section 5 shows various categories of optimization methods. Different categories are illustrated in Section 5, Section 6, Section 7, Section 8, Section 9 and Section 10. Section 11 compares the present work with the existing survey, and Section 12 concludes the work while highlighting the challenges.

## 2. Motivating Use Cases of Edge AI

Real-time analysis and decision making without latency—In contrast to cloud Al, edge AI can provide many benefits to the healthcare domain. For example, consider an individual is taking up his routine exercising. His vital parameters, namely pulse rate, blood glucose level, and blood pressure, are being monitored by wearable sensors and got updated in the edge application running on his mobile. Here, the mobile is the edge device, and it analyses the monitored parameters with the help of an artificial intelligence-based model, and it immediately takes the decision according to the analysis done, without sending the data to any other central server such as a cloud. The emphasis is that without any latency, the data are analyzed locally, and the decision is taken immediately. This is more important in the case of the healthcare domain, as the vital parameters are out of the normal threshold ranges. In such instances, immediately, the edge application intimates the physician and books the ambulance to a hospital.

Edge AI in Remote robotics surgery—In medical exigency, robotics surgery would be carried out under the supervision of a surgeon from remote. In this situation, the robots are fully equipped with AI-based models, and it performs the concerned surgery with guidance and conversation with a remote physician. The key point to be noted here is that the evolving 5G communication makes surgery easy and safe.

Edge AI integrated into cameras in airport security systems—When the video cameras installed in airports are integrated with video analytics applications, say, for example, detection of terrorist attacks, the attacks can be detected without latency. In conventional security applications, a series of video cameras capture the video of what is happening in the airport to cloud servers where AI-based models would run to detect or predict terrorist attacks. When the video analytics models run in the camera itself, based on the seriousness of the analysis, the device (i.e., video camera) makes a call to the police immediately to avoid the escape of any detected terrorist.

Edge AI in predictive maintenance—In manufacturing and in similar other sectors, predictive maintenance is used to determine potential faults and abnormalities in processes. Conventionally, heartbeat signals which exhibit the healthiness of various sensors and machines are continuously collected and sent to a centralized cloud setup where AI-based analytics would be carried out to predict the faults. However, nowadays, the loT devices themselves are equipped with AI models to predict the possibility of faults, and immediately with no latency proper maintenance process will be carried which ultimately leads to increased production with reduced cost.

Edge AI in quality assurance—Existing cameras and devices are incorporated with intelligent state-of-art neural network-based video analytics models, which are capable of executing Trillions of Operations Per Second. These devices have higher computer vision and scan a single product or huge batches of products at a time and find out faulty products with accuracy exceeding human capability. In addition to product inspection, edge devices are also equipped with relevant models for continuous and detailed factory monitoring. In addition, the complete assembly line of factories is thoroughly inspected by the installed cameras towards production with zero defects as shown in Figure 1.

Edge AI and mobile augmented reality—Mobile augmented reality applications have to process huge amounts of data that arrive from various devices such as a camera, OPS, and other video and audio data within the stipulated latency of around 15 to 20 milliseconds in order to show the augmented reality to the user [9]. The next generation IoT devices are equipped with reinforcement learning-based Al models implemented with the help of customized processors and 50 communication technologies. The combination of edge Al and the next-generation IoT enables practical implementation and brings in a more autonomous nature along with real-time analysis of the edge devices themselves. This combination will not only reduce the latency but also conserves energy.

Edge AI in drone applications—In contrast to traditional PID-based drone flights, which have limitations on the number of parameters and hence tend to break more often under situations that were not taken into the design, deep neural networks are trained with several example situations and simulated situations consisting of several disturbances, the AI-powered drone flight becomes easier. AT-powered drones are used for many applications such as (i) delivery of medicine to COVID-19 infected cases, (ii) delivery of food packets to affected people during natural disasters such as storms, (iii) tracking of vehicles, (iv) traffic monitoring, (v) air, noise, pollution monitoring, (vi) monitoring and exploring dangerous areas.

## 3. Review Method

Publications relevant to the objective of the paper have been collected from different data sources, namely Springer, IEEE, Elsevier, PubMed, Scopus, arXiv, MDPI, Hindawi, IEEE Access, etc., by using keyword querying method using the Google search engine. The resulting data are called the first dataset. Grey literature and publications irrelevant to the current objective have been excluded from the first dataset. This gives a second dataset. Another search has been performed from the cited publications of the second dataset, which would be combined with the final dataset. The above steps have been iterated for different keywords, including “optimization methods for edge AI”, “AI optimization methods for edge computing”, “AI model optimization techniques for edge computing”, “specialized hardware for edge computing”, etc., as shown in Figure 2.

## 4. Categorization of Optimization Methods

The details of the publications retrieved are shown in Table 1. Publications that lie outside the research question of the present work have been eliminated. i.e., the publications which have a broad scope and are relevant to general edge computing or machine learning, or general deep learning architectures and do not deal with the optimization aspect have been excluded. A final data set consisting of 107 publications has been taken up for analysis.

Short overview of the publications considered for the study is given in Table 2. The broad category of the publications along with their specific focus are given in Table 2.

With careful manual scanning of the publications, they are categorized as given in Figure 3.

In addition, the above categorization can be visualized, as in Figure 4, by placing it against the corresponding layers of the inherent technology stack, which is inherently associated with any AI-based application. The relevant publications fall into majors and subcategories, as shown in Figure 5.

## 5. Hardware Optimization

The hardware layer is the core layer for optimization because it only decides the computer, memory, storage, and networking capabilities of an application. The compute capability of an application is decided by the processor. Traditional processors were developed as general purpose Central Processing Units (CPU) which are primarily designed for sequential programming with few cores. CPUs have the highest Floating Point Operations per Second and large memory capacity. CPUs work based on the fetch-and-execute model according to Von Neumann architecture, wherein an instruction fetch cannot occur at the same time as an execution of data operation, causing instructions to be executed sequentially. Although in multi-core processors, the above concept is forbidden by sharing the data among several cores, the latency got increased. CPU processors are suitable for training small models with small batch datasets. However, the computation time of CPUs becomes prohibitively large for large dimensional data. Major producers of CPUs include Intel, AMO, Samsung, IBM, HP, etc.

Applications such as object recognition and text classification involve high-dimensional data. In addition, the neural network architectures also evolved with an increased number of layers to perform the task at hand with minimal error. Accuracy in applications such as image recognition and speech recognition has been achieved with Graphical Processing Unit (GPU) at the cost of huge resources and high energy consumption. A GPU has thousands of processors, and it breaks down complex problem into several tasks and solves them in parallel. With the help of parallel computing on its thousands of millions of processors, GPUs can perform graphics processing, video processing, machine learning, matrix computations, etc., with high throughput. GPUs are faster than CPUs. For example, from comparative performance analysis on a system for the classification of web pages using a Recurrent Neural Network as described in [10], the performance of the Tesla k80 GPU is found to be 4–5 times faster than the Intel Xeon Gold 6126 CPU. In GPU, parallelism is brought through software with two different techniques, Single Instruction Multiple Threads (SIMT) and Single Instruction Multiple Data (SIMD). GPUs are programmed in languages such as CUDA and OpenCL and hence provide limited flexibility when compared to CPUs. Major producers of GPU include Nvidia, ARM, and Broadcom.

Though GPUs serve as a better choice for executing complex AI models, recent applications are driven by the loT, and implementing AI solutions with GPU becomes infeasible in IoT devices [11,12]. Specialized hardware designs using Field Programmable Gate Array (FPGA), Application Specific Integrated Circuit (ASIC), Digital Signal Processor (DSP) [13], and Tensor Processing Units (TPU) have opened up alternate hardware optimization techniques for implementing AI models in edge devices. Field Programmable Gate Array (FPGA) is used to develop customized architecture for specific applications and thus gives high performance at low cost and low power consumption when compared to CPU and GPU. FPGA consists of up to millions of logic blocks, memory cells, and specialized 1/0 and Digital Signal Processing (DSP) blocks which can be programmed to implement a specific function in terms of logical operations. Typically, the required application has to be coded in Register Transfer Language (RTL), which will then be compiled by RTL compilation tools such as Synplify from Synopsys. The output is called netlist, which represents the application in terms of connections between the logic blocks. The netlist passes through two more steps called placement and routing, which appropriately implement the connections required on the hardware. The steps involved in the FPGA workflow are shown in Figure 6.

Another desirable feature of FPGA is that it is reconfigurable so that a variety of applications can be implemented on FPGA [14]. The reconfiguration is at first stored in an off-chip non-volatile memory, and it becomes effective during its write to static RAM [15].

Application Specific Integrated Circuit (ASIC) refers to more specialized hardware designed for a specific purpose. When compared to FPGA, ASIC is not reprogrammable. However, ASIC is more energy efficient than FPGA. In addition, the size of ASIC is smaller when compared to FPGA. However, it consumes time to design ASIC. It is comparatively expensive. Different types of ASIC include Tensor Processing Unit (TPU), Vision Processing Unit (VPU), and Neural Processing Unit (NPU).

TPU is an Application Specific Integrated Circuit specially designed for matrix computation with extreme parallelism and with high throughput via open-source machine learning software such as Tensorflow. It is specially optimized for the deep neural network. Producers of TPU are Google, Coral, and Hailo. Further, the evolution of TPUs can be visualized in four generations [16]. The first-generation TPU is a CISC processor, and the complex instructions are executed by Matrix Multiplier Unit. This TPU works on 8-bit integer operations, and this kind of TPU is more suitable for inference. The second-generation TPU works on floating point operations. The second-generation TPU can be used to implement both model training and model inferencing. Further, the third-generation TPU is more powerful than its predecessors and has high throughput. An important aspect of TPC is that the unit will not follow the fetch and execute model as CPU. However, the CPU based host will fetch the instruction and load it in TPU for execution. Its focus is fully on matrix computations involved in the neural network. Another key point to be noted is that the above-said TPUs are typically used in cloud infrastructure, whereas the edge TPU (4th generation TPU) are designed specifically for edge devices where the neural network-based models are run on the top of light weight machine learning frameworks such as TensorFlow Lite.

VPU facilitates computer vision-based inferences in edge devices with high power efficiency without compromising high performance. For example, Intel’s Neural Computing Stick 2 (NCS 2) is optimized by minimizing the data movements using programmable computation strategies on a workload-specific hardware accelerator, Intel’s Movidius Myriad X VPU. Vision processing units are designed specifically to process visual or image data acquired by edge devices such as cameras. These low power consuming (around 2 to 3 watts) units are typically programmed for object recognition and facial recognition applications implemented with the Google TensorFlow framework.

NPU is having data-driven parallel computing architecture designed to process massive multimedia data such as video and images for specific applications. NPU architecture is dedicated to energy-efficient deep neural networks [17] having diverse hardware-software co-optimization schemes for inference with low power consumption. The small form factor of NPU makes it useful in many applications deployed in cell phones, such as smart security cameras, gesture-controlled autonomous drones, and industrial machine vision equipment. For a quick reference, the power consumption, prediction error, and throughput are different processors compared in Table 3.

## 6. Learning-Related Strategies

### 6.1. Federated Learning

Federated learning is a distributed, and collaborative learning method that allows different edge devices with different datasets to work together to train a global model. In this learning, a single global model is stored in a centralized cloud infrastructure. At first, the global model is shared with devices with initial weights. Now the edge device collects the real-time data and trains the model locally with the new data for one or several iterations in order to update the weights so that the loss function is minimized [19]. The updated weights are sent to a centralized server. Here the data are not sent to the centralized server. Only the weights are sent to the server with encryption [4]. The centralized server receives the updated weights from several edge devices. It computes the average of updated weights, and then it updates the weights of the global model. Then the global model is again shared with edge devices. The concept of federated learning is shown in Figure 7. Federated learning could cater to the needs of modern IoT-based applications and turns out to be the basis for next-generation artificial intelligence [20].

As discussed above, in centralized learning, the server sends a model, which initially gets trained in the server, to edge devices. In an edge device, the model gets training with local data, and the updated model parameters are sent to the server. In this way, the server aggregates all the parameters and resends the updated model to devices. In the case of decentralized, federated learning such as Peer-to-Peer [22] or in a ring topology [23], in addition to training the model with local data, each device performs the updating of parameter values with a Gossip algorithm [24,25]. As multiple devices are involved in training the model simultaneously, the training time is reduced [26]. Moreover, in federated computing, privacy and security of data are maintained [27]. Federated learning is more appropriate for utilizing low-costing machine learning models on devices and sensors [28].

As far as the data in different clients are homogeneous in which the feature space of local data in participating clients is the same, updating the global model with updated weights of clients would be easy, and such data federated learning is called horizontal learning, and it leads to an effective global solution [29]. However, updating the global models becomes a challenge when the data in devices are of the kind non-Independent and Identically Distributed (non-IID) which makes the convergence of the global model becomes difficult [30]. This issue can be resolved by developing a personalized local model for each client [31], and then the personalized local models can be merged into a shared global model with the help of the Bayesian fusion rule, as discussed in [32,33]. In another research work [34], a branch-wise averaging-based aggregation method has been proposed, which guarantees convergence of the global model. In another work [35], feature-oriented regulation method has been proposed to establish a firm structure information alignment across collaborative models.

Federated learning is of two kinds, centralized [21] and decentralized (Figure 8).

### 6.2. Deep Transfer Learning (DTL)

Edge devices such as IoT, webcam, robots, intelligent medical equipment, etc., are very useful for many healthcare applications during a pandemic, say, for example, COVID-19. Both shortages of reliable datasets, limited hardware, and power support of edge devices prohibit the usage of deep learning models in them. However, in Deep Transfer Learning (DTL), the knowledge of an already learned model is used to solve a new task, as in Figure 9. DTL significantly reduces training time and the requirement of resources for a target domain-specific task for a fixed feature extraction or fine-tuning [36,37].

In contrast to conventional machine learning, where learning takes place in isolation, transfer learning uses knowledge learned from other existing domains while learning for a new task. Transfer learning is of two types, homogeneous and heterogeneous. In homogeneous transfer learning, the source and target domains are the same. It means that the feature space is the same, whereas, in heterogeneous transfer learning, the feature space is not the same. Different methods, namely, instance-based methods, feature-based methods, parameter-based methods, and relation-based methods, are being used for homogeneous transfer learning, whereas for heterogeneous domains, only feature-based methods are being used. There are two types of methods, namely asymmetric feature transformation and symmetric in heterogeneous transfer learning. In asymmetric transfer learning, one of the domains is transformed into the other, and this method is found to be effective when both domains share the same label space. In symmetric transfer leaning, both domains are transformed into a common feature space.

### 6.3. Knowledge Distillation

Large machine learning models have millions of parameters associated with them, which makes the deployment of the model in edge devices infeasible. So, the knowledge gained from large models is transferred to small models which run on edge devices. Here, the large models serve as the teacher model, and the small model is such as the student model. The teacher model refers to a larger model, and it alone gets pre-trained. The learned knowledge from the teacher model is transferred to the student model through knowledge distillation, as in Figure 10. The knowledge distillation helps in improving the accuracy of the student model despite the constrained hardware [38]. Different knowledge distillation techniques, namely response-based distillation, where the prediction performance of output layers of the teacher model and student model are compared using a loss function which shows the difference between the models and the loss function is minimized so that the accuracy of student model approaches that of the teacher model. In contrast to response-based distillation, in feature-based distillation teacher model distills the intermediate features of the student model. Here, the position of distillation is moved prior to the output layer [39,40]. Here the student mimics by minimizing the loss function that is computed according to the intermediate layers. Relation based distillation is one in which the difference between the relationship between different feature maps is captured as a gram matrix, and the corresponding loss function is minimized. The student model reduces the computation cost and memory usage [41].

In addition, knowledge distillation is also used to tackle the problems, namely, heterogeneity in the local data of different devices and heterogeneity in the architecture of the models in edge devices. Federated learning can be implemented through knowledge distillation, as discussed in [42,43].

## 7. Model Optimization Methods

In the fields of computer vision, natural language processing, video analytics, etc., deep neural networks are being used in innovative and impactful ways. However, the computational resources required for implementing neural networks are on the higher side. Further on, the energy consumption of these architectures is also high, while the heat dissipated by them into the environment is hugely damaging. Deploying such features-rich DNNs in the Internet of Things (IoT) edge devices is beset with a number of technical challenges and concerns due to the limited hardware and power resources in edge devices. Thus, DNN models ought to be optimized in order to substantially reduce the usage of huge computational resources.

### 7.1. Pruning

Larger models require more memory and more energy, are hard to distribute, and consume long computation time. Pruning is used to produce models having a smaller size for inference. With reduced size, the model becomes both memory efficient and energy efficient and faster at inference with minimal loss. Pruning is implemented by removing unimportant connections or neurons, as in Figure 11.

Pruning is of two kinds. Unstructured pruning simply removes neurons, whereas structured pruning removes neurons and their respective connections, weights, and channels [44]. Weights with very small values may be pruned. For example, in [45], the authors reduced the number of parameters of AlexNet from 61 million to 6.7 million and of VGG-16 from 138 million to 10.3 million, both with no accuracy loss by pruning unimportant connections along with fine tuning of weights. Some of the parameters in the network are redundant and do not contribute anything great to arrive at the output. When one computes the rank of the neurons in a network according to how much they contribute, then the low-ranking neurons can be removed from the network. This results in a smaller and faster network. In Ref. [46], the layer-wise pruning method has been proposed in which two neurons produce highly correlated outputs, then the outputs are pruned to one, and the error due to the pruning of the neuron is corrected by the method of least square. One key point to be noted here is that the removal of neurons or connections or weights based on their magnitude will lead to changes in the structure of deep neural networks. In contrast to such pruning, filter-based pruning will achieve acceleration but not by changing the structure of the deep neural network, which get supported by commercially available off-the-shelf deep learning libraries, as discussed in [47]. In this work, the authors have obtained a ‘thin network’ by pruning the unimportant filters. Moreover, pruning of a filter in the ith layer leads to the removal of the corresponding channel in the (I + 1)th layer, as in Figure 12. So, filter pruning and channel pruning [48] are correlated [49]. Pruning can be evaluated in terms of model size, accuracy, and computation time.

### 7.2. Quantization

Fundamentally, quantization refers to reducing the precision of weights, parameters, biases, and activations so that they occupy less memory and the size of the model would be reduced. In an artificial neural network, weights are 32-bit floating-point values. Consider a neural network with millions of parameters. Here, the memory requirement to store millions of 32-bit floating point values is too high to get accommodated in edge devices. So, the 32-bit floating point values are converted into typically 8-bit integers. During quantization, the range of parameters or weights must be scaled to the 8-bit integer range (i.e., −127 to +127). This process is called scale quantization. In addition to scale quantization, the data for quantization has to be grouped, called quantization granularity. I.e., whether the quantization is going to be applied per channel (in 3D input) or per row, or per column (2D input).

In Ref. [50], the authors demonstrated that the accuracy of the deep neural network is not influenced when the 32-bit floating point values of weights are trimmed to 16-bit fixed point numbers when image classification has been performed using the MNIST database [51] and CIFAR10 database [52]. In other research work such as [53,54], where the layer-wise weights and inner products are approximated from 32-bit floating point values are quantized into very low binary values, say −1 and +1, and then the quantization error is minimized by comparing the original and quantized models. Towards correction, the proximal Newton algorithm has been proposed in [55] and an alternate method [56] where the authors encode the loss difference due to approximation in the loss function by associating explicit loss-aware quantization with an incremental strategy, as discussed in [57]. In this incremental strategy, the layer weights are split into two portions, weights of one portion are first approximated to low bit values, whereas the weights of the other portion are kept with the original 32-bit floating bit number and retrained to minimize the quantization loss due to the first portion. The groupwise quantization and retraining are iterated to minimize the loss in accuracy.

In general, the quantization process is employed after training the model (called post-training quantization). Then weights and activations of the trained models are quantized and embedded into the model. Now, this model has to be evaluated for its accuracy, as quantization is associated with inherent accuracy loss. The accuracy loss can be resolved using different methods such as partial quantization, quantization-aware training, and learning quantization parameters. In partial quantization, quantization is employed only for a few layers so that the accuracy of the model is maintained. Similarly, during post-training quantization method, the accuracy of the model may not be sufficient enough for the given task. The model with quantized weights is retrained or tuned to tune the accuracy to the desired level, as shown in Figure 13. Similarly, during training, different quantization parameters, namely scale value, range value, and zero point extra, are learned, which will be used to fine-tune the model for its required accuracy.

### 7.3. Weight Sharing

As discussed in [58], there is much redundancy among the weights of a neural network, and this proves that a small number of weights are enough to reconstruct a whole network. In the weight-sharing method, the number of effective weights that are required to be stored is reduced by having multiple connections in a neural network share the same weight [59]. The weight sharing may be based on a random method or HashedNet in which the weights are grouped according to a Hash function [60,61]. The concept of weight sharing is exemplified in Figure 14.

Consider a layer in the neural network that has 4 input neurons and 4 output neurons. The weight matrix of this layer is 4×4. There are 16 weights, and each weight is represented using 32 bits. Now the number of bits required to store the 16 weights is 16 × 32 = 512 bits. Now, weight sharing groups the weights in buckets or clusters share the same value. Consider that the 16 weights share 4 shared values (in the example, there are 16 weight values, 1,2,3 up to 16. Now, consider that the 16 values are grouped into 4 clusters or groups, for example, group-1 (1,2,3,4), group-2 (5,6,7,8), group-3 (9,10,11,12), and group-4 (13,14,15,16). Now, consider that each weight is replaced by the group average. Then the shared weights would be 2.5, 6.5, 10.5, and 14.5. Moreover, at least two-bit indexing is required to point to the 4 shared weight values, as shown in Figure 14. Then, the original weight values are replaced by the indexes of the shared weights, and the shared weights are stored with 32-bit representation. Here, to represent 4 shared weight values, 2-bit indexes are required. Now, the number of bits involved in this example is 4 × 32 + 16 × 2 = 160 bits. In addition to weight sharing, the shared weights may further be compressed using Huffman coding [62] during storage.

### 7.4. Matrix Decomposition

Matrix decomposition is a mathematical technique in which complex matrix operations are performed on the constituent matrices (decomposed matrices) rather than on the original matrix. There are various methods. Namely, Lower Upper (LU) decomposition, which is used to decompose square matrices, QR decomposition, which is used to decompose any rectangular matrices and Cholesky decomposition is for square symmetric matrices. Matrix decomposition is also called matrix factorization, and it gives a more compact representation of a full matrix in hand. Matrix factorization reduces the number of operations on a full matrix, say A of dimension (m × n) where ‘m’ denotes the number of rows, ‘n’ denotes the number of columns, into a decomposed representation UV^T^, where the matrix U having dimension (m × d) and matrix V has dimension (n × d). Here, ‘d’ denotes the embedding dimension. With this factorization, it is understood that the number of operations with full matrix A, say, O(nm) reduced to O((n + m) d), and d is much smaller than m and n. Thus, matrix factorization helps in reducing the size of the model at hand.

In Ref. [63], matrix decomposition has been employed for the weight matrix between the hidden and output layer with the intention to arrive at low-rank weight matrix, which in turn speeds up the performance of the neural network. In contrast to the above work where the matrix factorization is employed in the hidden to output later, in [64], the authors employed the matrix factorization method in the first forbidden layer with an intension to produce low-rank weight matrix, which helps in reducing the model parameters significantly. In addition, the authors could achieve a 75% reduction in parameters without any loss in performance.

### 7.5. Stochastic Gradient Descent (SGD)

In a neural network, the goal of gradient descent is to minimize the cost function, which represents the difference (i.e., error value) between the predicted value and the actual value of the output by updating the values of weights and bias. This method takes the initial values of the parameters and updates the values with calculus-based operation such that the cost function becomes minimum. Two components are involved in gradient descent optimization, namely, cost function and learning rate. The learning rate is the step size with which the parameters are updated. If it is too large, it may overlook the minimum. If it is too small, it may require many iterations to reach the minimum value for the cost function. There are different kinds of gradient descent algorithms being used, including batch gradient descent, stochastic gradient descent, and mini-batch gradient descent. Batch gradient descent sums the error for each data in the training dataset and updates the parameters of the model only after determining the error values of all the data in the dataset. In the stochastic gradient descent method, the method updates the weights and biases after finding out the error value for each data item. Since this method involves only one data item, the memory requirement for this optimization approach is small and fits as a natural choice for deploying neural network models in edge devices [65]. In contrast to the above two methods, the mini-batch gradient method, method splits the training dataset into several mini-batches and updates the weights and biases after each mini-batch.

### 7.6. Gradient Scaling in Network Quantization

As mentioned earlier, in a deep neural network, weights and biases are typically quantized from their 32-bit floating point representation to 8-bit fixed format representation. Moreover, further acceleration of the models is being obtained using still lower 4-bit or 2-bit binary formats in which multiplication and addition operations use XNOR and bit count operations. Despite the quantization of weights and bias in the forward pass of the neural network, more training time is consumed in the GEneral Matrix Multiplication (GEMM) computations of the backward pass of the neural network. In order to make the GEMM efficient, the full length 32-bit floating point representation is quantized into 4-bit floating point representations. This reduces memory space and enhances performance.

In Ref. [66], the authors have scaled gradient values into 4-bit floating point representations, in addition to the 4-bit integer quantization of weights and bias parameters. With the proposed method, the have obtained a significant acceleration greater than 7× over the existing 16-bit floating point representation. This novel method enabling 4-bit training of deep learning models can be used across many domains.

In Ref. [67], an elementwise gradient scaling technique is used, which adaptively scales up or down the given gradient using Hessian information for backpropagation and achieves more stability and accuracy of the model.

### 7.7. Regularization

When a deep neural network model is developed for a purpose, underfitting or overfitting issues may arise. Overfitting of the model occurs when the developed model learns more about the training data, and it works very well with training, but it will not produce the same performance when the new unseen dataset is given. This is because the model learns too much about the particulars of training data. Regularization is a technique used to resolve the issue of overfitting by reducing the magnitude of features.

There are different regularization techniques, L1 regularization (Lasso), L2 regularization (Ridge), and dropout. Consider that the cost function of a model is represented as given in (1)
(1)cost_function=loss+regularization_term

Here, the loss refers to the difference between the actual and predicted values for a single input data item, whereas the cost function refers to the sum of loss for all the data items present in the entire training dataset.

For the L1 technique, the cost function is expressed as in (2)
(2)$$\cos t=loss+\lambda \times \sum_{i=1}^{n}\left|w_{i}\right|$$

Here, λ is called the regularization parameter and λ>0. This regularization is used to reduce the complexity of a model. The key point with the L1 regularization technique is that the technique takes absolute weights, which implies that a weight value can take zero value. That is, some of the features can be completely removed. Thus, the L1 regularization technique is used for both feature selection and reducing overfitting issues.

In the L2 technique, the cost function is expressed as in (3)
(3)$$\cos t=loss+\lambda \times \sum_{i=1}^{n}w_{i}^{2}$$

Here, the cost function is altered by a penalty term which is computed by multiplying λ by the sum of the squared weight of the individual features. Here, the penalty term regularizes the coefficients of the model and thereby reduces the complexity of the model. Both in Ll and L2, the final value of $w_{i}$ is not only influenced by the model and data but also by a predefined parameter λ which is independent of the model and data. Thus, we can prevent overfitting if we set an appropriate value of λ, though too large a value will cause the model to be severely under fitted.

Drop-out regularization is another technique employed on any non-output layer of a deep neural network to handle overfitting issues. In this method, during training, some layer outputs are dropped out. A term called threshold retention probability is used to specify the probability that a neuron is not dropped. For example, if the threshold retention probability is set as, say, 0.9, if a neuron has a retention probability that is less than 0.9, then it will be dropped. In addition, retention values for neurons are set using a random generator. It means that the neurons are dropped at random. This technique is specifically used during the training of a model in order to make the model more robust against fluctuations in the training data. So, when some neurons are dropped out during training, then during testing or inference, there would exist more connections in the succeeding layer, i.e., neurons in the subsequent layer get more excited or more activated, which also should be compensated.

## 8. Hyperparameter Tuning

In machine learning, there are two types of parameters, namely model parameters, and hyperparameters. In a deep neural network, weight and bias refer to model parameters that are computed from training data, whereas a hyperparameter, say, the learning rate is a hyperparameter that will be tuned to obtain the model’s optimal performance. Different hyperparameter tuning methods are classified in Figure 15.

Tuning hyperparameters becomes necessary [68] both in the complex model, where many hyperparameters are required to be tuned, and in a lightweight model, each parameter has to be tuned strictly to a particular range. Training-related hyperparameters, such as the batch size of the data and learning rate, influence the speed of convergence, and structured-related hyperparameters decide the model learning capacity.

In general, hyperparameter optimization can be formulated as in (4).
(4)$$x^{*}=\min f(x)$$

In (4) f(x) denote the objective function used for determining optimal values for hyperparameters. It may be such as Root Mean Square Error (RMSE) expression, and it needs to be minimized. $x^{*}$ denotes the optimal values of the given hyperparameters, denoted by, say x. In addition, the objective function may be of maximizing type also.

Grid search performs an exhaustive search on the hyperparameter set and range given by a user so that the optimal values that yield the highest accuracy are obtained. Since the method works in a brute-force search manner, to include all possible combinations, the computing resources required for this search are heavy. When limited search space is given, this method becomes more appropriate. In contrast to the grid search method, the random search method determines an optimal set of hyperparameters by randomly selecting the values within the search space given. The random is typically performed till the desired accuracy is obtained. In addition, this method is adopted as a first-level optimization to find out the range of hyperparameters, and then they will be fine-tuned. Both grid search and random search suffer from one limitation the method does not consider the previously evaluated values when trying to find new values for the hyperparameter set.

In contrast to the grid and random searches, the Bayesian optimization method takes into consideration the previously calculated /past values while attempting to find the new values for hyperparameters. The Bayesian model, first, builds a surrogate probability model of the form p(y|x) where y denotes the probability score for the given hyperparameters, denoted by, say x. i.e., the surrogate objective function is a high dimensional mapping of hyperparameters to the probability of a score on the objective function. It finds the optimal x for which the score is maximum. Now, these values will be used in the true objective function. The results of the true objective function are updated to the surrogate, and the above process is repeated until the desired accuracy is obtained. Further, the method chooses the next set of hyperparameters with the help of the selection function, which is of the form, as shown in (5).
(5)$$f_{y^{*}}(x)=\int_{-\infty }^{y^{*}}\left(y^{*}-y\right)p(y|x)dy$$

Also, the Bayesian optimization method is formulated as Sequential Model-based Optimization. There are different types of surrogate functions, such as the Gaussian process, Random Forest Regression, and Tree Parzen Estimator (TPE). The Bayesian method finds optimized hyperparameters with fewer evaluations compared to gird or random search methods.

Gradient-based optimization is used to optimize several parameters simultaneously based on the computation of gradient with respect to hyperparameters [69]. In this work, the authors presented a method that simultaneously tunes the hyperparameters by backpropagating the gradient through Cholesky decomposition and back-substituting the model parameters. This has reduced the mathematical computation involved in tuning significantly to a measure of about $\left(s^{3}/3\right)$ operations in cases where the training criterion (which is continuous and differentiable with respect to model parameters and hyperparameters) is a linear function of model parameters such as linear regression.

In the population-based training method, a population of neural networks is trained using randomly selected hyperparameters simultaneously, and at the end of, say, ‘m’ iteration, the performance of the networks has been computed based on, say, for example, loss function. Then the hyper parameters of the best-performing network will be used by the population for further exploration of fine-tuning the hyperparameters. i.e., within the population, the training is not independent. A network is allowed to copy the hyper parameters of the best-performing network and to continue its further hyperparameter tuning. It this way, it is a combination of parallel search and sequential optimization [70,71].

With respect to tools for optimizing the hyperparameters of the models, the following are some of the important ones.

Scikit-learn has implementations for grid search and random search.Scikit-optimize uses a Sequential model-based optimization algorithm to find optimal solutions for hyperparameter search problems in less time.Different libraries, namely, BayesianOptimization, GPyOpt, and Hyperopt, are some of the implementations available in Python for optimizing hyperparameters of models using the Bayesian optimization method.
○BayesianOptimization tool implemented in Python. This tool finds out the optimal hyperparameters with two important functions, (i) objective function and f(x), in which x denote a set of hyperparameters used to optimize f(x). In addition, in most of the models, the inputs and outputs of the model are known. (ii) acquisition function, which is based on the Gaussian process model, which serves as a surrogate for f(x), used to determine new values of x.○GPyOpt is another Python implementation of Bayesian optimization used to automatically configure the models.○Hyperopt is a Python library for serial and parallel optimization over search spaces, designed with three algorithms, namely random search, TPE, and adaptive TPE.
Keras library is used to determine the optimized hyperparameters such as kernel size, filters, number of neurons in each layer, and learning rate for deep learning models from the given range of the hyperparameters. Internally, it uses random search and grid search methods.Ray Tune uses the power of distributed computing to speed up hyperparameters with different population-based training methods. In addition, it supports different machine learning frameworks such as TensorFlow, Sklearn, PyTorch, etc.Optuna—To facilitate the users while defining search space for hyperparameters in large-scale applications, Optuna provides an input facility that dynamically constructs the search space in contrast to other tools where the search space is statically given. The tool has random search, grid search, Bayesian optimization based on Gaussian process or Tree-structured Parzen Estimator and Covariance Matrix Adaption - Evolution Strategy based method, etc., for identifying the optimized values. Another unique feature of Optuna is that it provides different algorithms such as asynchronous successive halving algorithm, hyperband algorithm, median pruning algorithm, threshold pruning algorithm, etc., which allow the users to customize their requirements. The tool is installed in Python.Metric Optimization Engine (MOE) is more suitable when the objective function is a black box, the derivatives are unavailable, or any other internal knowledge is not available.Spearmint determines optimal hyperparameters using Bayesian optimization, and it automatically adjusts the number of parameters to minimize the objective function in fewer runs.Sigopt provides high customizable search spaces and multimetric optimization with sophisticated tuning options.

When hyperparameters of a model are optimized, there are chances that the tuned model may be affected by overfitting (the model learns too much from the training dataset, and the variance is high) of underfitting (the model has not learned enough from the dataset and the bias value is high) issues. In general, though the lowest bias and lowest variance are desired, one cannot be lowered without increasing the other, which is called a bias-variance trade-off. So, hyperparameters are tuned for an optimal trade-off between bias and variance. Again, the chosen optimal hyperparameters are required to be validated with the K-fold cross-validation method. Since K-fold cross-validation divides the dataset into K folds and uses data that belong to (K-1) number of folds for training the model with data of kth split as testing data, repeating the above procedure K times and finally averaging the model performance, the issues of overfitting and underfitting are getting eliminated. When hyperparameters are optimized for a given dataset, the model may over perform well only for the considered training data.

## 9. Energy Efficiency Techniques

Basically, power refers to the rate at which energy is consumed over time. In edge AI, energy per inference and the maximum number of inferences per second are used as energy and power metrics. In general, edge devices are battery-operated devices that store the total energy (capacity) in the battery, and the capacity of the battery is specified using a metric called mW-H (milliWatt Hours). For example, consider a 12 V battery with a specification of 100 AH (Ampere-Hour). Now the battery capacity is calculated as 12 V × 100 Ah = 1200 WH. It implies that the battery can deliver 1200 Watts in one hour before its discharge. Moreover, assume that the battery is giving power to an IoT device that consumes 2 W in one hour. Then the battery can power the device for 600 h. Power (in Watts) is expressed as Energy (in Joules) consumed in one sec. Power efficiency is measured as the unit of performance per unit of power consumed.

The power efficiency of an edge device that performs inference tasks is computed as the number of inferences per second (IPS) per Watt. Similarly, the power efficiency of another device that is used for, say, an image recognition application is computed as the number of Frames per Second (FPS) per Watt. The power efficiency of a processor can be measured as Terra Operations per Second (TOPS) per Watt. While designing neural network-based applications for edge devices, the supply voltage and clock speed of the processors should be taken into account. The power consumed is directly proportional to the square of the supply voltage. The speed of the application needs to be optimized without necessarily increasing the supply voltage.

Secondly, as far as energy and power optimization methods are concerned, edge devices should be capable of consuming less energy for inference tasks. Moreover, the inferencing task should be carried out only when it is actually required. Here, the power consumption for performing maximum inferences/second is computed as given in (6).
(6)$$P_{max-inferences}=\left(Maximum\_number\_of\_inferences/second\right)\times \left(Energy/inference\right)$$

Thirdly, as mentioned above, the inferencing task should be performed only when it is actually required. So, there would be another power term, called static power which is required to keep the edge device in normal functioning mode. So, the total power required would be as in (7)
(7)$$\begin{array}{c} P_{total}=P_{static}+P_{max-inference} \end{array}$$

### 9.1. Neural Architecture Search (NAS) and Hardware-Aware NAS

NAS is one of the model compression techniques used for computer vision tasks [72,73]. Basically, NAS is an algorithm-based technique used to find an optimal neural network architecture automatically to resolve the problem at hand with good accuracy, minimum time, and cost. The technique has three entities in it, search space, search strategy, and performance estimation. The search strategy selects an architecture from the search space which contains many possible architectures for the given problem in hand (here, human intervention is involved in inputting the relevant architectures) and determines its performance. Basically, the evaluation of the model involves the application of the model on a dataset for its training and validation [74]. The major advantage of NAS is that it takes small datasets and, with limited search, space helps to obtain a better model in a quick time. In the earlier NAS, the method automatically searches for an efficient deep neural network architecture for the given problem but without taking into account the hardware characteristics while finding out the DNN architecture, whereas the hardware-aware NAS searches for efficient DNN architecture, taking into account the hardware characteristics, say for example GPU [75].

### 9.2. Algorithm-Accelerator Co-Design Method

In this method, AI algorithm designers and hardware developers work together to determine the between optimization solutions. In this approach, both algorithm design alternatives and accelerator design alternatives are tried simultaneously to achieve the optimized solution, which has been discussed for the first time in [76]. Further different, effective algorithm-accelerator co-design approaches have been discussed in [77]. The integrated approach of considering both algorithm and accelerator/hardware design will bear the biggest benefit in energy consumption. Consider a matrix operation of 16 × 16 dimensions. This needs a corresponding memory block of the same dimension in hardware also. Say in case the hardware dimension is not corresponding to 16 × 16; rather, it says 8 × 8, then more cycles are required for performing the matrix operations, which leads to inefficient implementation. So, if the hardware and software design are taken together, it will lead to optimized computing and memory solutions. Ultimately, energy consumption will also be optimized.

In addition, neural networks often depend on large vector-matrix multiplications, and hence analog accelerators are well suited for this task. New hybrid chip designs, combining analog accelerators and conventional digital accelerators, will allow us to perform operations automatically on the particular processor that’s most suitable for the task. The co-optimization of innovations in data usage, hardware, and software will bear the biggest benefit in energy consumption. To create truly energy-efficient AI systems, we thus need an integrated approach that tunes these innovations.

### 9.3. Memory Optimization

Toward providing energy-efficient mechanisms for edge AI, on one side, AI developers are making efforts to optimize the models through various mechanisms such as pruning, quantization, etc. On the other side, hardware development advances with modern circuit design even beyond Moore’s Law. Despite the above efforts, still, power consumptions need to be optimized as the number of IoT devices keeps on evolving very rapidly. In edge AI, memory is the critical component that consumes more power. First, the full DNN models do not get accommodated into the on-chip memory as their size is limited. So, the system has to access off-chip DRAM access which has latency and energy requirements higher than other computer operations [78]. To reduce DRAM access, layer partitioning and scheduling has been discussed in [79]. The access time and energy spent for typical DRAM access will vary based on the requested access met with row buffer hit (means that the requested row is available in row buffer), row buffer miss (means that there is no activated row in the buffer) or row buffer conflict (there is a row in the buffer but which is different from the requested one).

Typically, DRAM consists of a channel, rank, chip, bank, row, and column, with each bank, consisting of multiple sub-arrays, each sub-array local row buffer. The latency in memory access can be reduced by reducing the row buffer miss and row buffer conflict or by increasing the row buffer hit. In Ref. [80], the memory latency has been reduced by using a DRAM data Mapping policy (DRMap) which achieves maximum row buffer hits while exploiting bank-level parallelism and sub-array level parallelism. In Ref. [81], the authors have discussed sub-array level parallelism in the same bank in three different DRAM architectures, namely, SALP-1(in this architecture, pre-charging of sub-array is combined for parallelism with activation of another sub-array), SALP-2(here, write-recovery latency (which refers the time that must be elapsed between last write command to row and pre-charging of it) is used for overlapping the activation of another sub-array) and SALP-MASA(activation of multiple sub-arrays is performed simultaneously).

In addition, in another research work [82], layering partitioning and scheduling have been proposed to minimize DRAM access. This method is based on the fact that CNN processes the data sequentially in a layerwise manner, i.e., one layer at a time. Moreover, the CNN processing of data in a particular layer can be perceived into two loops, namely inner and outer, whereas the inner loop refers to the processing of a particular portion of data that is currently in the on-chip, and the outer loop refers to the processing of the subsequent portions of the data. Further, the data are partitioned into blocks that are of the same size as on-chip memory. The outer loops are subsequently scheduled, and the number of loops refers to the number of partitions. The outer loops are scheduled according to the sequence. Here, whenever the subsequent operations share the same data, then the data present in the on-chip memory itself will be reused for the subsequent operation and thereby reducing the memory accesses. In another research work [83], an adaptive scheduling algorithm has been proposed that provides not only the dynamic partitioning of data and its reuse but also considers the constraints of adjacent layers.

### 9.4. Energy-Efficient Communication Protocols

In general, performing an AI task on edge involves the transfer of information between edge AI devices and edge service in multiple rounds. Federated learning is typically employed in edge devices due to the fact that centralized learning is infeasible in edge computing devices which have the inherent difficulty of limited computing power and storage. Even with federated learning in which edges need to work only with their local data, different optimization techniques are being used. In order to enhance the efficiency of federated learning, first and second-order methods have been used in cases where gradient information is available. In case the stochastic gradient is not available, the zero methods is being used. In first-order learning, the loss function in any edge device is of the form given in (8)
(8)$$loss\_function,F(w)=\frac{1}{n}\sum_{i=1}^{n}f(w_{i},x_{i})$$
where n denotes the number of samples available in the devices. Now, in each edge device, the updated weight matrix is computed using the gradient of the loss functions, which is of the form given in (9)
(9)$$W_{k+1}=W_{k}-\eta \nabla F(w)$$

Here, η denotes the learning rate and ∇F(w) denotes the gradient of the loss function.

But the gradient method has the drawback of a slow convergence rate. It implies that the communication of local gradients to the server needs to be carried out in multiple rounds.

Computation and communication are heavy. Whereas the second order derivative methods, which are of the form given in (10)
(10)$$W_{k+1}=W_{k}-\eta {\left(\nabla^{2}F(w)\right)}^{-1}\nabla F(w)$$

Here, ${\left(\nabla^{2}F(w)\right)}^{-1}$ denote the inverse Hessian matrices (say $H^{-1}$). This second order derivative is the Newton method, and it required only a few iterations to obtain an updated weight matrix. In addition, the size of Hessian matrices is large, and it is also optimized using approximate inverse Hessian matrices [84,85]. Further, in case the gradient of the loss function is not available in an edge AI-based application, then Zero order derivation, only functional value will be used for updating the weight matrix [86]. In zeroth-order methods, only functions are evaluated and not their gradients.

In addition to the above kind of optimizing computation and communication of information, there are other optimization techniques that look into routing communication in a manner that makes the transfer energy efficient, as discussed in [87,88,89]. Bluetooth Low Energy (BLE) can provide wireless communications distances between 10 and 100 m [90]. Wi-Fi Hallow offers low power consumption (comparable with Bluetooth) while maintaining high data rates and a wider coverage range [91]. ZigBee was conceived to provide low energy consumption by being asleep most of the time and waking up periodically. Moreover, it is easy to scale Zigbee networks and create mesh networks to extend the IoT node communications range. Long-Range Wide Area Network technology is used to deploy Wide Area IoT networks with low energy consumption. Even a cluster head may be selected under diverse IoT devices of wireless sensor networks for reliable and energy-efficient data transmission [92].

### 9.5. Communication Efficiency with Gradient Compression

Despite the implementation low power wireless protocols such as Bluetooth, Wi-Fi, etc., edge devices are of weak communication capabilities. As mentioned earlier, in federated learning, the edge devices compute the gradient and send the values to the server. In order to enhance the weak communication efficiency of devices, different methods have been proposed to enhance the exchange efficiency. Either quantizing gradients to lower bits or communication, the gradients which are greater than a given threshold will enhance the communication efficiency [93]. In Ref. [94], the authors proposed a method to reasonably select gradient values based on the information entropy of gradient distribution which is regular and almost normal and filters the communication of gradient according to the selection.

In Ref. [95], a method has been proposed for an efficient exchange of gradient in distributed deep learning, where the transfer of gradient is delayed until an unambiguous gradient having high amplitude and low variance is computed. In addition, the authors found that the method had a high compression ratio while maintaining accuracy. In another research work [96], the authors proposed a method for gradient compression in which gradients are considered as random variables which are distributed according to some sparsity-inducing distributions, and then they compressed the gradient based on a threshold scheme that enables deep gradient compression while having lower computation overhead and higher speed.

### 9.6. Gradient Checkpointing

In Refs. [97,98], a method called Gradient Check Pointing (GCP) has been used to facilitate the training of edge devices having low memory with the image or video databases whose input size and features are very large. With this method, during the forward pass, only a subset of matrices/tensors called gradient checkpoints are stored in the memory. During the backward pass, the required or missing tensors are computed with the required reforward passes.

### 9.7. Shared Memory Concept

The dense convolutional architecture, in its naïve implementation, requires huge memory [99]. Densenet involves a large number of concatenations and batch normalization operations. The values are required to be stored contiguously to make the operations efficient. This requires huge memory. In memory efficient implementation of densenet, two techniques have been followed. (i) since the concatenation and batch normalization operations do not consume much computation time, they have been recomputed whenever they are required. The gradient also may be recomputed whenever it is required. (ii) the memory space is shared for storing the outputs of concatenations. Similarly, in contrast to deep networks, in an inception network, speed and accuracy are enhanced by using filters of different sizes (say, for example, 1×1, 3×3 and 5×5) at the same level, which makes the network wider rather than deeper. Then the outputs of different filters are concatenated after max pooling and then given to the next inception layer. Secondly, implementing filters of size N×N as (1×N) and (N×1).

Normally, in the deep neural network, convolutional operations are performed over all the input channels. For example, consider an input of size ($D_{f}\times D_{f}$) and M channels. Consider a filter of size $D_{k}\times D_{k}$ is employed over all M channels, then the number of multiplication operations in one channel would be $D_{k}\times D_{k}$, and for M channels, it would be $D_{k}\times D_{k}\times M$. Now the filter is moved over the input features both horizontally and vertically, say for $D_{p}$ times, then the number of multiplications would become $D_{k}\times D_{k}\times M\times D_{p}\times D_{p}$. Moreover, if there are N filters, then the number of multiplications in a normal CNN is $D_{k}\times D_{k}\times M\times D_{p}\times D_{p}\times N$.

In contrast to normal CNN operations, in mobile net, operations are performed in two steps. Firstly, the depth-wise separable convolution operation is formed to only one channel at a time, and secondly, pointwise convolutional operations are performed following the depthwise operation.

In depth-wise convolution, the operation is performed over one channel at a time, and the kernel size is $D_{k}\times D_{k}\times 1$. Now the filter is moved over the input features both horizontally and vertically, say for $D_{p}$ times, then the number of multiplications would become $D_{k}\times D_{k}\times D_{p}\times D_{p}$. Now for M channels, the number of operations would be $D_{k}\times D_{k}\times D_{p}\times D_{p}\times M$. After depthwise convolution operation, with the filter size 1×1 and for M channels, the filter size is 1×1×M. N such filters are required. When this filter is applied for $D_{p}$ times both horizontally and vertically, the number of operations would be $M\times D_{p}\times D_{p}\times N$. The *total_number_of_operations* and *reduction_ratio* are obtained as given in (11) and (12) [100]
(11)$$total\_number\_operations=D_{k}\times D_{k}\times D_{p}\times D_{p}\times M+M\times D_{p}\times D_{p}\times N$$
(12)$$reduction\_ratio=\frac{D_{p}\times D_{p}\times M(D_{k}\times D_{k}+N)}{M\times D_{p}\times D_{p}\times D_{k}\times D_{k}\times N}=\frac{1}{N}+\frac{1}{D_{k}\times D_{k}}$$

## 10. Delineating the AI Model Optimization Framework

In the software industry, different frameworks are being widely used for significantly moderating development, deployment, and management complexities. Software engineers and architects have been leveraging many enabling frameworks in order to speed up software production to streamline and simplify business fulfillment. In short, any complicated activity is substantially made simple through the smart usage of appropriate frameworks. The same is true in the context of edge AI also. In the recent past, the process of AI model generation has been picking up as there are several sophisticated business tasks getting resolved through pioneering AI models. However, producing path-breaking AI models is neither straightforward nor simple. AI model engineering is definitely a complicated activity. Going forward, AI models are being evaluated, retrained, optimized, deployed, and observed to keep up their performance in production environments. In short, AI model engineering comprises many tough tasks. However, brute-force AI model production results in verbose AI models. That is, the computational, memory, storage, and network bandwidth requirements go up considerably with heavyweight AI models. For training, testing, and retraining purposes, AI models consume a lot of IT resources. It is crucial to have a facilitating framework for optimization methods for edge AI with the following components

AI-specific processing unitsAI Model Optimization Techniques (Pruning, Quantization, etc.)Architecture-centric Learning Techniques (centralized and decentralized architectures)Hybrid Approaches (Federated learning is fused with knowledge distillation, etc.)Hyperparameter Optimization (HPO) Tuning ToolsetsEnergy-efficiency Methods

This paper has detailed each of the above enablers to enlighten our esteemed readers. Based on the problem at hand and the target fixed, the way forward is being facilitated through this facilitating framework.

## 11. Related Survey Works

There are some survey works related to optimization methods of edge AI. However, the perspective and focus of the survey vary considerably. For example, the research work [101] conducted an extensive survey of an end-edge-cloud orchestrated architecture for flexible AI of Things. The research work [102] discusses the confluence of edge computing and artificial intelligence by segmenting it into two categories, viz., intelligence-enabled edge computing and artificial intelligence on edge. Similarly, the research work [11] has a specific focus on hardware and software optimization for DNN. The survey [4] explores the optimization methods for edge networks. The survey [3] provides an overview of recent progresses in machine learning in the domain of IoT. In Ref. [7], the authors performed a survey with a specific focus on AI hardware platforms and software packages. In Ref. [40], a more focused survey on knowledge distillation has been carried out. In Ref. [103], the authors reviewed the motivation for AI running at the edge and discussed architectures, frameworks, and emerging key technologies for deep learning models towards training and inferencing at the edge. The works [19,20,25,26] presented a survey with a specific focus on federated learning. Optimization methods for scheduling computation offloading are reviewed in [104]. In contrast to the above survey, the present survey presents an elaborate survey on optimization methods for edge AI with a holistic view.

## 12. Conclusions

To optimally run AI models in the IoT edge devices in order to fulfill the long-pending demand of realizing real-time insights, state-of-the-art AI model optimization techniques are emerging and evolving fast. In this paper, the authors have identified and illustrated the proven and potential methods with the aim of making edge devices equipped with artificial intelligence-based models. The combating techniques for surmounting the challenges and concerns associated with edge AI implementation are discussed in this paper. In addition to AI model optimization approaches such as pruning, quantization, knowledge distillation, and their combination, there are endeavors for bringing the much-required optimization through hardware accelerators and AI-specific processing units. Federated learning through centralized and decentralized architectural patterns and styles is being presented as a viable model optimization technique. Finally, hyperparameter optimization (HPO) tuning methods and toolsets have been flourishing in the recent past as a way of bringing deeper and more decisive automation in fine-tuning AI models. As widely known, compression is a double-edged weapon. Thus, enabling compression without compromising model performance is the need of the hour. The need for a facilitating framework to implement the required optimization method in a customized manner according to the application at hand is highlighted.

Through this survey, it is ascertained that edge devices can be equipped with artificial intelligence to deliver low latency and reliable functions at low power consumption. Despite the advancements in edge AI, real challenges are associated with it its implementation. First, the cost is one of the major challenges associated with the implementation of edge AI. IoT technology requires fast and revolutionary innovation to bring down the cost associated with the design, development, installation, and usage of IoT sensors [105]. Another important challenge is with respect to security and privacy. Lightweight, reliable encryption techniques are required for edge devices, and such solutions should be efficient enough to be integrated with an edge, 5G, and other emerging technologies [106]. Bringing ubiquity in edge AI is really a challenge. AI-based edge solutions should be made very flexible as well as plug and play so the users are not required to be technologically literate. Moreover, the data in edge devices are highly heterogeneous, and integration of data is a big challenge. In the future, edge devices will be the key enabler for sixth-generation communication networks [107] where the latency is less than 1 ms. Edge and edge AI have to meet the global coverage needs of 6G. These challenges create future directions of research.

## Figures and Tables

**Figure 1 sensors-23-01279-f001:**
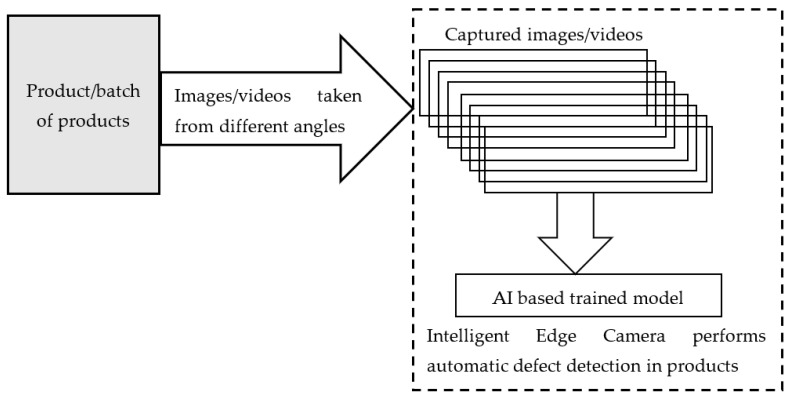
Block diagram of edge AI-based defect detection.

**Figure 2 sensors-23-01279-f002:**
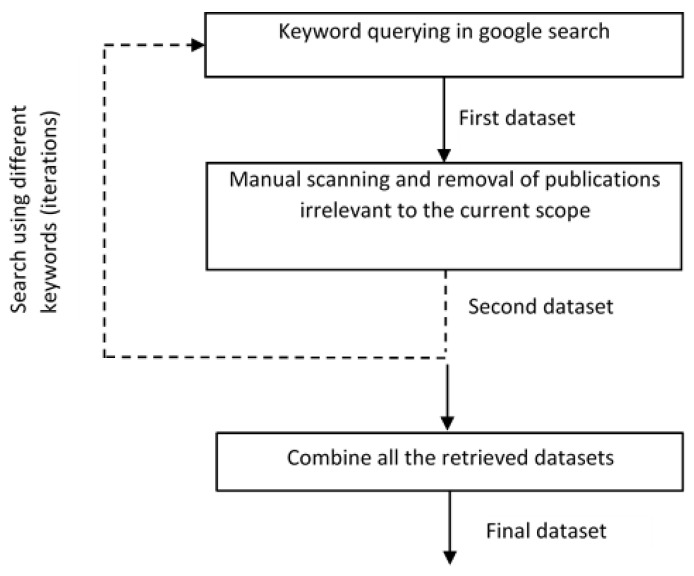
Review method.

**Figure 3 sensors-23-01279-f003:**
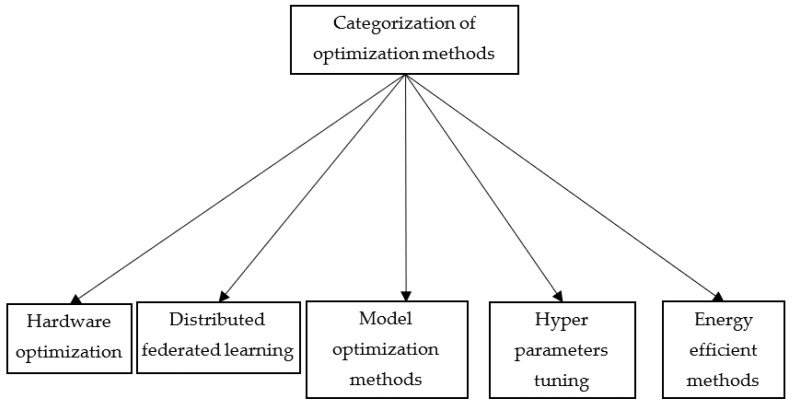
Categories of optimization methods.

**Figure 4 sensors-23-01279-f004:**
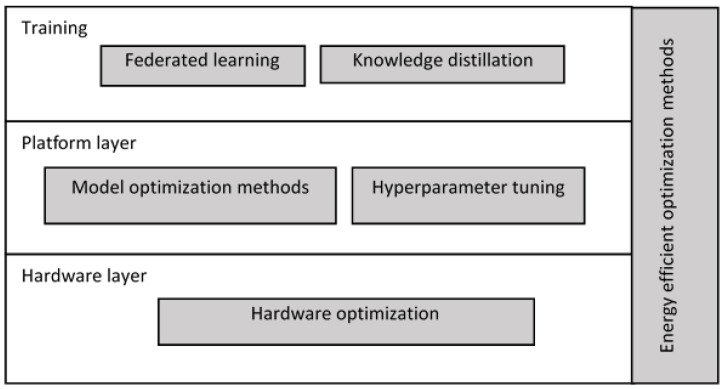
Categories of optimization methods are placed against the inherent technology stack of an AI application.

**Figure 5 sensors-23-01279-f005:**
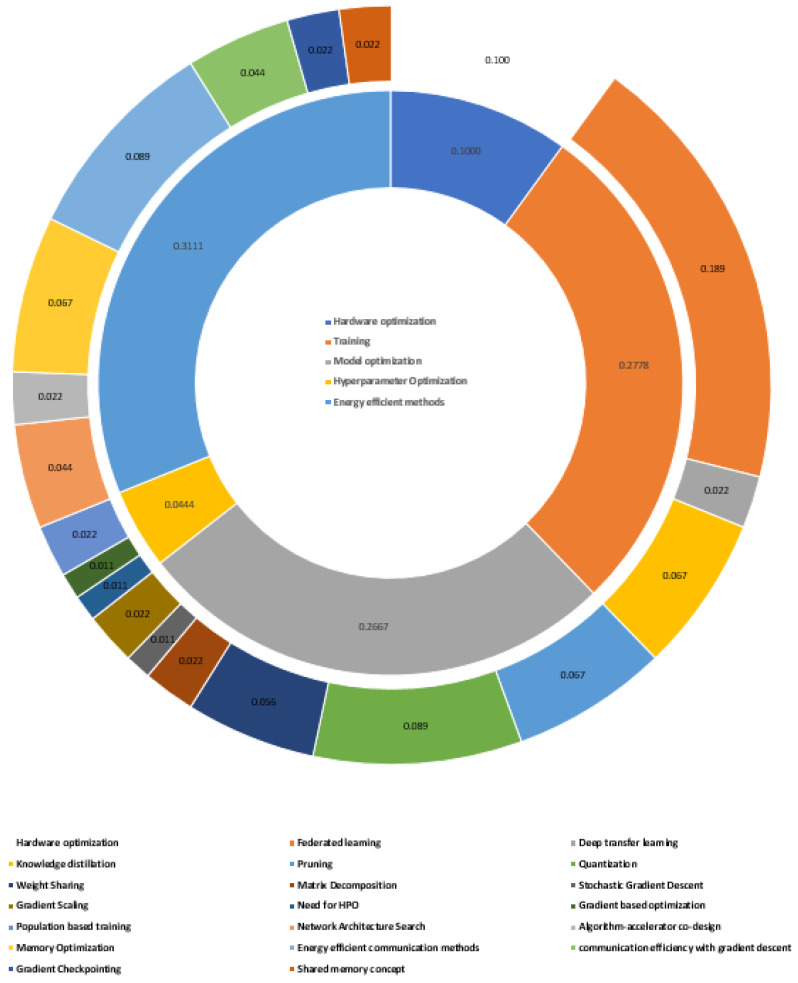
Graph denoting the major and sub categories of publications.

**Figure 6 sensors-23-01279-f006:**
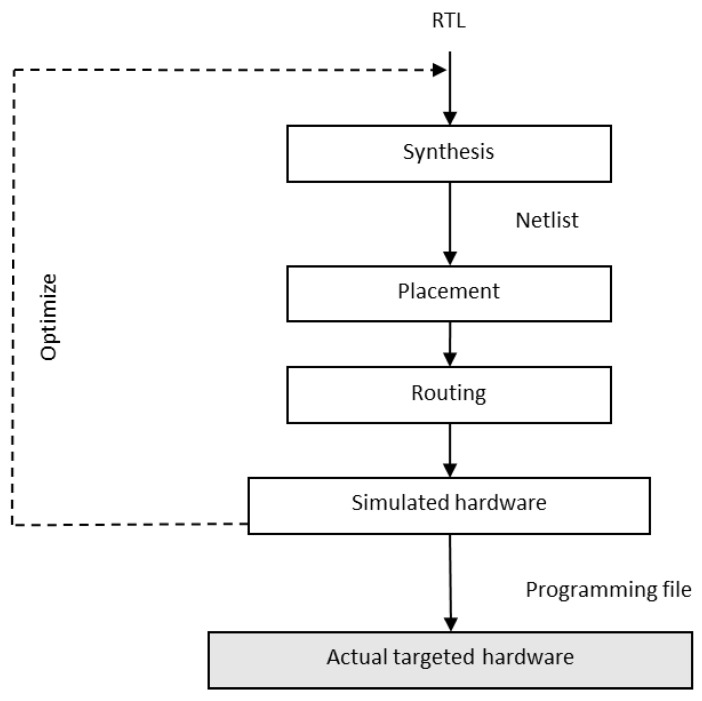
Steps in FPGA.

**Figure 7 sensors-23-01279-f007:**
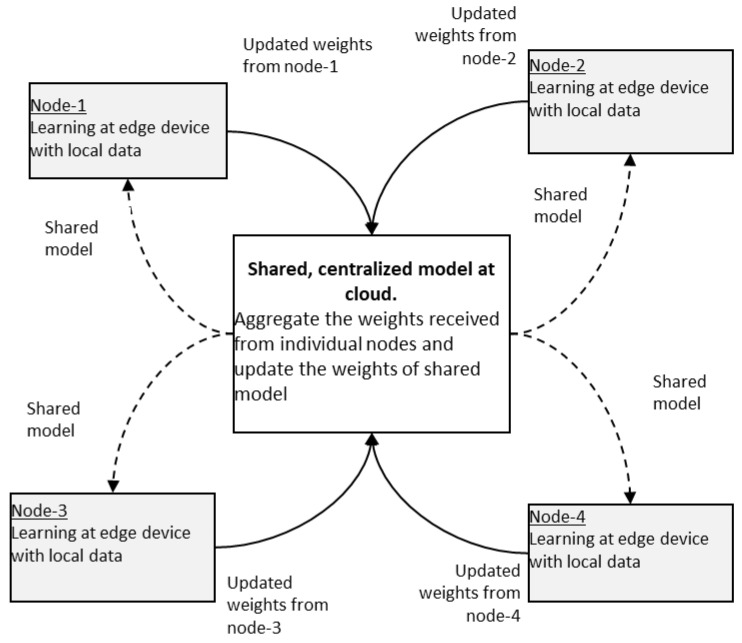
Concept of federated learning in edge servers (shown with four nodes).

**Figure 8 sensors-23-01279-f008:**
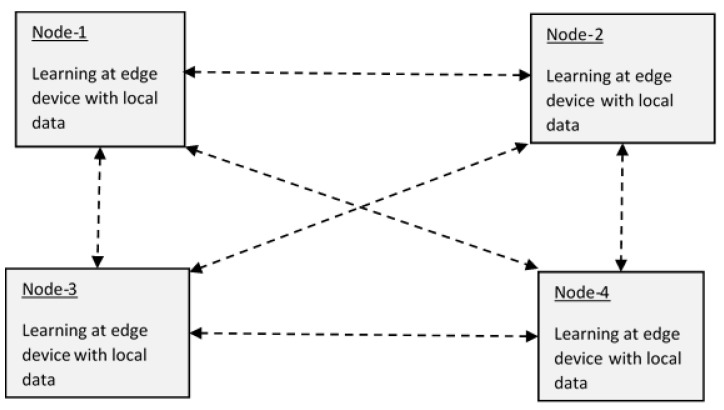
Federated learning in a peer-to-peer model.

**Figure 9 sensors-23-01279-f009:**
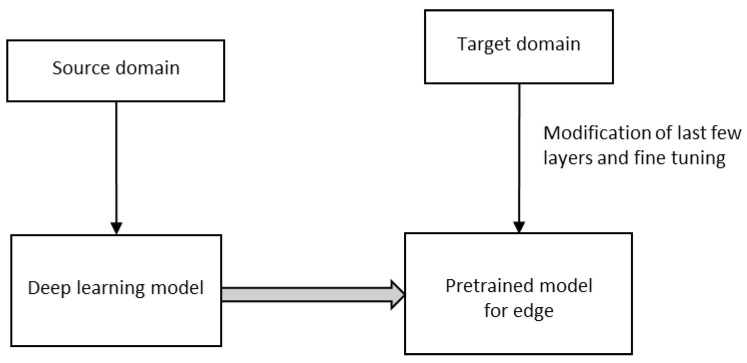
Deep transfer learning.

**Figure 10 sensors-23-01279-f010:**
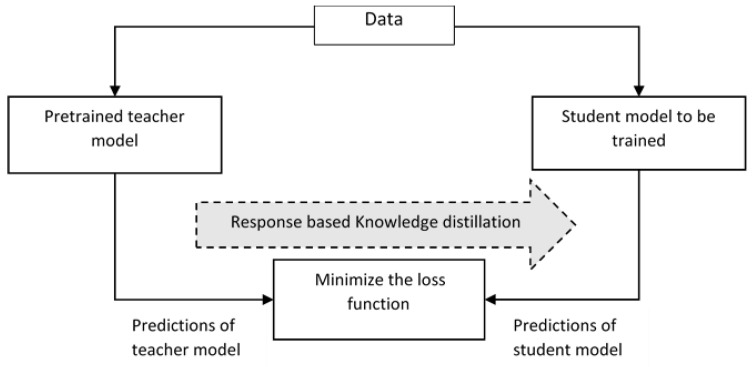
Knowledge distillation.

**Figure 11 sensors-23-01279-f011:**
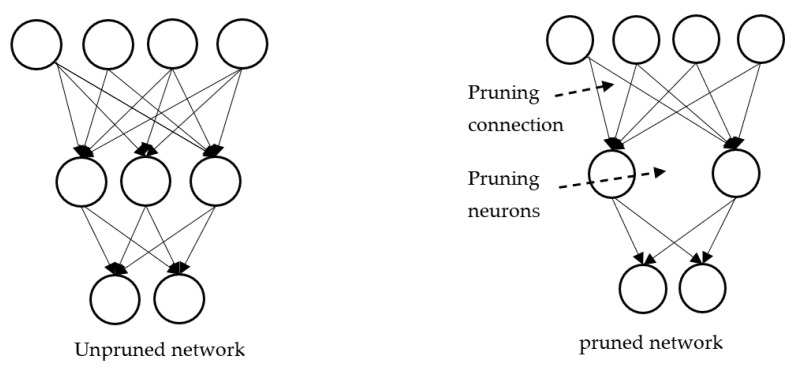
Pruning of connections and neurons.

**Figure 12 sensors-23-01279-f012:**
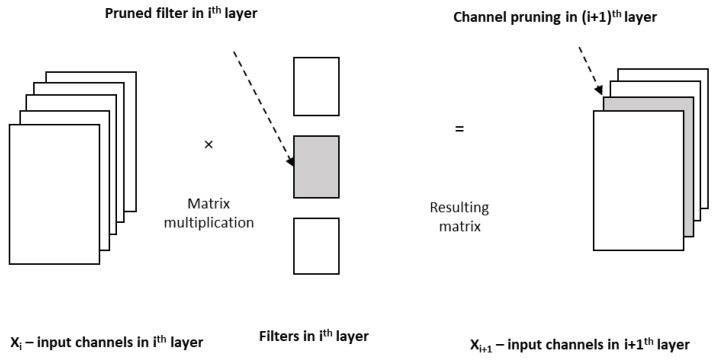
Filter and channel pruning.

**Figure 13 sensors-23-01279-f013:**
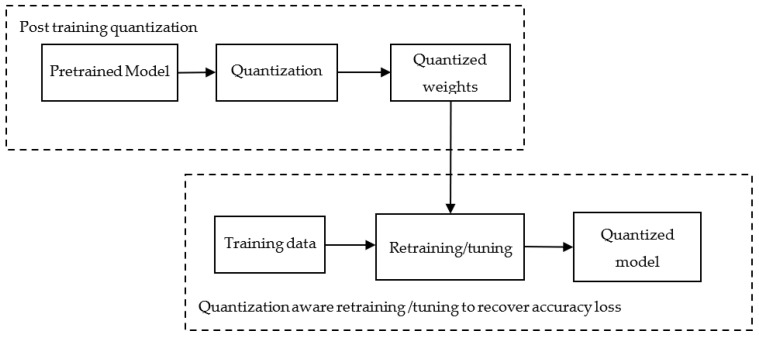
Post-training quantization and quantization-aware retraining.

**Figure 14 sensors-23-01279-f014:**
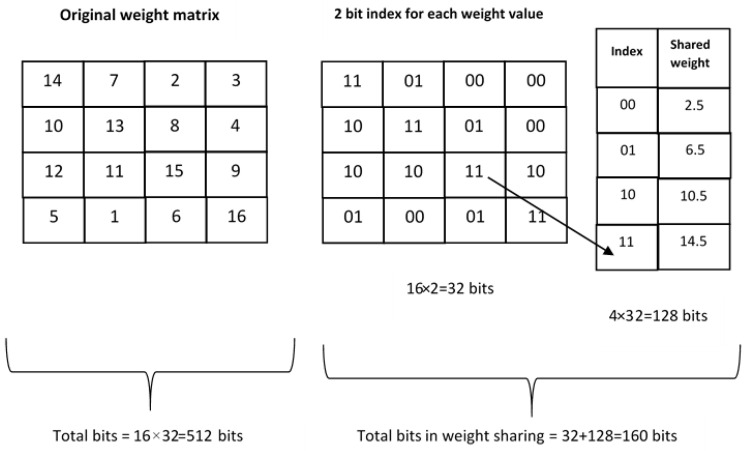
Weight sharing.

**Figure 15 sensors-23-01279-f015:**
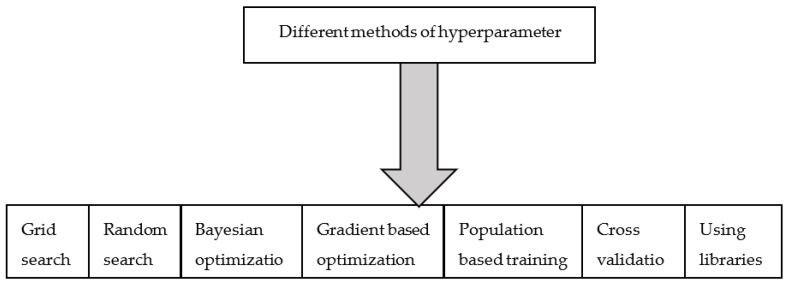
Different Hyperparameter tuning methods.

**Table 1 sensors-23-01279-t001:** Details of publications retrieved.

S. No.	Description	Count
1	Total number of publications collected from different electronic databases	129
2	Publication with broad scope related to generic edge computing publications	9
3	Publications related to machine learning concept but not dealt with edge	7
4	Publications related to general DNN architecture and not handled optimization	6
5	Number of publications relevant to the current scope	107

**Table 2 sensors-23-01279-t002:** Short overview of the relevant publications.

References	Broad Category	Specific Focus
[1,2,3,4]	Need for edge computing	Emphasize the need for processing at edge devices to meet the needs of applications which cannot tolerate latency, high bandwidth and transmission energy related issues. Emphasize the need for real time analysis data in applications such as autonomous cars, video analytics, streaming applications, applications having augmented reality, remote robotic based applications, etc.
[5,6,7,8]	Challenges in equipping edge devices with AI	Describe the hardware and energy constraints associated with edge devices
[9]	Edge AI use cases	
[10,11,12,13,14,15,16,17,18]	Hardware optimization	Deep neural network models have become inevitable in many applications which involve face recognition, object detection, image classification, etc. Specialized processors and accelerators are mandatory for creating, training and deploying deep learning models in edge devices The publications discuss about the evolution of processors and accelerators including Field Programmable Gate Array (FPGA) and Application Specific Integrated Circuit (ASIC).
[19,20,21,22,23,24,25,26,27,28,29,30,31,32,33,34,35]	Training	Discuss about distributed federated learning, its types, issues such as non-Independent Identically Distribution of data and how to resolve such issues
[36,37]		Discuss about the difficulty in the direct implementation of deep learning models and propose deep transfer learning as an alternate
[38,39,40,41,42,43]		To enable training with small data sets in edge devices, knowledge distillation method is discussed
[44,45,46,47,48,49]	Model Optimization	Discuss about different levels of pruning namely neuron pruning, connection pruning, filter pruning, channel pruning in deep neural network towards compressing the model size
[50,51,52,53,54,55,56,57]		Discuss about the quantization of 32 bit floating bit weight into 8 bit integers and also about low bit quantization
[58,59,60,61,62]		Sharing of weights with mapped level so that number of bits to be stored would be reduced
[63,64]		Discuss about matrix decomposition where a method of working with low ranked matrices rather than the high dimensional original matrix
[65]		While computing gradient descent, stochastic gradient descent method is employed to reduce the size
[66,67]		Discuss about gradient scaling where the 32 bit gradient values are scaled to short integers
[68]	Hyper parameter tuning	Need for Hyperparameter Optimization(HPO) for achieving efficient models
[69]		Hyperparameter tuning through Gradient based optimization
[70,71]		HPO tuning through population based training
[72,73,74,75]	Energy efficient methods	Network Architecture Search
[76,77]		Algorithm-accelerator code sign
[78,79,80,81,82,83]		Memory optimization
[84,85,86,87,88,89,90,91,92]		Energy efficient communication methods
[93,94,95,96]		Communication efficiency with gradient descent
[97,98]		Gradient checkpointing
[99,100]		Shared memory concept
[101,102,103,104,105,106,107]	Review publications	Reviews on edge intelligence and challenges

**Table 3 sensors-23-01279-t003:** Comparison of power consumption, prediction error, and throughput of different processing units.

Type of Processing Unit	Power Consumption (in Watts)	Prediction Error in (%)	Throughput (in Giga Operations per Second (GOPS))
GPU	>10^2^	<1	10^3^ Giga Operations Per Second
CPU	10^2^	Around 1	1 to 10 Giga Operations Per Second
FPGA	<10	>1 and ≤10	10 to 100 Giga Operations Per Second
ASIC	<1	≥10^1^	10 to 100 Giga Operations Per Second
Edge TPU	** 2 W	* Accuracy is 0.77% when compared to 96% of GPU	4 trillion operations per second
VPU	<1 W	*** 32% with throughput 3 times higher	over one trillion operations per second
NCU	<1 W	**** Quantization error 0.81%	11 trillion operations per second

* https://www.springml.com/blog/machine-learning-on-tensor-processing-unit/ (accessed on 30 December 2022); ** https://cloud.google.com/tpu/docs/tpus (accessed on 30 December 2022); *** https://viso.ai/edge-ai/vision-processing-unit-vpu-for-inference/ (accessed on 30 December 2022); **** [18].

## Data Availability

Not applicable.

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
