# Peer review of "A Survey on Optimization Techniques for Edge Artificial Intelligence (AI)"

_sensors, 2023, doi:10.3390/s23031279_

Round 1

Reviewer 1 Report

+ present a summary consistent with the research relevant to the field.

+ The introduction needs improvement, avoid so many examples with one example is enough, preferably the one of the medical parts and the one of the autonomous cars I recommend to remove it, it goes a little out of context instead of explaining it confuses also it is good that the justification that is implicit in the introduction contains a little more references alluding to the problem.

+ improve future work

Author Response

+ present a summary consistent with the research relevant to the field.

Response

Done

+ The introduction needs improvement, avoid so many examples with one example is enough, preferably the one of the medical parts and the one of the autonomous cars I recommend to remove it, it goes a little out of context instead of explaining it confuses also it is good that the justification that is implicit in the introduction contains a little more references alluding to the problem.

Response

Yes, autonomous car example has been removed.  Introduction section has been improved

+ improve future work

Response

The challenges are highlighted in the conclusion which create avenues for future work

Reviewer 2 Report

Horizontally distributed inference of deep neural networks for AI‐enabled IoT

The review provides an in‐depth discussion of the most salient approaches conceived along those lines, elaborating on the most pertinent aspects concerning the partitioning schemes exploited and the parallelism paradigms explored, discussing in an organized and schematic manner the underlying workflows and associated communication patterns, as well as those DNNsʹ  features at both the macro‐  and micro‐architectural level that have guided the design of such techniques, also highlighting the primary challenges encountered at the design and operational levels as well as the specific adjustments or enhancements explored in response to them.

However some minor changes have to be adopted

In introduction the authors can come up with the existing survey works on the similar topic.

Table1.RecentresearcheffortsthatstudythedistributionofDLworkloadswithinanIoTcluster provides very good information

A MapReduce‐like distributed programming model is used to coordinate CNN inference computations in a synchronized fashion, across a given number of mobile and embedded devices. Mention the importance of Mapreduce

The authors can come up the challenges faced

Add the lessons learnt

The main contribution of the survey lies in challenges and future directions these are missing

The authors can refer A hybrid cluster head selection model for Internet of Things, An Evolutionary Secure Energy Efficient Routing Protocol in Internet of Things

Author Response

Reviewer-2 comments

Horizontally distributed inference of deep neural networks for AI‐enabled IoT

The review provides an in‐depth discussion of the most salient approaches conceived along those lines, elaborating on the most pertinent aspects concerning the partitioning schemes exploited and the parallelism paradigms explored, discussing in an organized and schematic manner the underlying workflows and associated communication patterns, as well as those DNNsʹ  features at both the macro‐  and micro‐architectural level that have guided the design of such techniques, also highlighting the primary challenges encountered at the design and operational levels as well as the specific adjustments or enhancements explored in response to them.

 However some minor changes have to be adopted

In introduction the authors can come up with the existing survey works on the similar topic.

Response

Existing survey papers included in section 11

Table1.RecentresearcheffortsthatstudythedistributionofDLworkloadswithinanIoTcluster provides very good information

 A MapReduce‐like distributed programming model is used to coordinate CNN inference computations in a synchronized fashion, across a given number of mobile and embedded devices. Mention the importance of Mapreduce

Response

The focus of the paper very specific to edge AI where Mapreduce models could be employed. 

The authors can come up the challenges faced

Add the lessons learnt

The main contribution of the survey lies in challenges and future directions these are missing

Response

Included in Section 12

The authors can refer A hybrid cluster head selection model for Internet of Things, An Evolutionary Secure Energy Efficient Routing Protocol in Internet of Things

Response

The above reference included

Reviewer 3 Report

The authors try to review the optimizations for AI Edge and  the concept is good. The manusciprt can be modified with a few suggestions as 

1) Short the keywords

2) Correct the references 

3) Number the equations and present equations in a proper manner

4) Explain the concept of Figure 2 and Table 1

5) Make a proper linkage with technologies reviewed to make it better understandable to readers

Author Response

Reviewer-3 comments

The authors try to review the optimizations for AI Edge and the concept is good. The manuscript can be modified with a few suggestions as 

1) Short the keywords

Response

 Done

2) Correct the references 

Response

 Done

3) Number the equations and present equations in a proper manner

Response

 Equations are numbered

4) Explain the concept of Figure 2 and Table 1

Response

 Explanation given

5) Make a proper linkage with technologies reviewed to make it better understandable to readers

Response

 Done